# A microRNA that controls the emergence of embryonic movement

**Jonathan AC Menzies, André Maia Chagas, Tom Baden, Claudio R Alonso\***

Department of Neuroscience, Sussex Neuroscience, School of Life Sciences, University of Sussex, Brighton, United Kingdom

**Abstract** Movement is a key feature of animal systems, yet its embryonic origins are not fully understood. Here, we investigate the genetic basis underlying the embryonic onset of movement in *Drosophila* focusing on the role played by small non-coding RNAs (microRNAs, miRNAs). To this end, we first develop a quantitative behavioural pipeline capable of tracking embryonic movement in large populations of fly embryos, and using this system, discover that the *Drosophila* miRNA *miR-2b-1* plays a role in the emergence of movement. Through the combination of spectral analysis of embryonic motor patterns, cell sorting and RNA in situs, genetic reconstitution tests, and neural optical imaging we define that miR-2b-1 influences the emergence of embryonic movement by exerting actions in the developing nervous system. Furthermore, through the combination of bioinformatics coupled to genetic manipulation of miRNA expression and phenocopy tests we identify a previously uncharacterised (but evolutionarily conserved) chloride channel encoding gene – which we term <u>Mo</u>vement <u>Modula</u>tor (*Motor*) – as a genetic target that mechanistically links *miR-2b-1* to the onset of movement. Cell-specific genetic reconstitution of *miR-2b-1* expression in a null miRNA mutant background, followed by behavioural assays and target gene analyses, suggest that *miR-2b-1* affects the emergence of movement through effects in sensory elements of the embryonic circuitry, rather than in the motor domain. Our work thus reports the first miRNA system capable of regulating embryonic movement, suggesting that other miRNAs are likely to play a role in this key developmental process in *Drosophila* as well as in other species.

**\*For correspondence:**
c.alonso@sussex.ac.uk

**Competing interest:** The authors declare that no competing interests exist.

## eLife assessment

This **important** study presents a new quantitative imaging pipeline that describes with high temporal precision and throughput the movements of late-stage *Drosophila* embryos, a critical moment when motion first appears. A new approach is used to explore the role of miRNAs in motion onset and presents **solid** evidence that shows a role for miR-2b-1 and its target Motor in embryonic motion. The data are well supported even if the mechanistic insight into the emergence of movement remains to be explored.

## Introduction

Movement is the main output of the nervous system allowing animals to walk, fly, crawl, swim and maintain their posture, so that they can find prey, mate partners, escape predators and relocate within habitats (*Biewener et al., 2022*). Despite the key biological and adaptive roles of movement across animal systems, how developing embryos manage to organise the necessary molecular, cellular, and physiological processes to initiate patterned movement is still unknown (*Hassinan et al., 2024*). Although it is clear that the genetic system plays a role, how genes control the formation, maturation, and function of the cellular networks underlying the emergence of motor control systems remains poorly understood.

Recent work in our laboratory has shown that miRNAs – which are short regulatory non-coding RNAs that repress the expression of target genes (*Alonso, 2012*; *Bartel, 2018*) – have pervasive roles in the articulation of complex movement sequences such as those involved in body posture control in the young *Drosophila* larva (*Issa et al., 2019*; *Klann et al., 2021*; *Picao-Osorio et al., 2015*; *Picao-Osorio et al., 2017*); these observations, as well as those from others in *Drosophila* and other systems (*Fricke et al., 2014*; *Holm et al., 2022*; *Kadener et al., 2009*; *Lackinger et al., 2019*; *Weng et al., 2013*) hinted at the possibility that miRNAs might also be involved in the control of more fundamental aspects of motor development and control.

In this study, we first investigate the impact of miRNA regulation on *Drosophila* larval movement and discover that the miRNA *miR-2b-1* is necessary for normal locomotion in freshly hatched first instar larvae. Based on this finding and the fact that, to a great degree, the biological properties of young early larvae are defined in the embryo, we hypothesised that *miR-2b-1* affects the emergence of movement during the embryonic stage. To test this possibility, we developed a novel behavioural pipeline capable to detect the first manifestations of embryonic movement, and using this new approach, we established that *miR-2b-1* is indeed essential for the normal development of embryonic movement through a role in the sensory nervous system. Through the combination of bioinformatic miRNA target prediction, gene expression and phenocopy analyses we identified a previously uncharacterised gene predicted to encode a chloride channel – which we call Motor – as a genetic link between miR-2b-1 and its effects on embryonic movement. Our findings suggest that other miRNAs are likely to play roles in the emergence of embryonic movement in *Drosophila* and other animal species.

## Results and discussion
### A miRNA that impacts larval movement

To explore the possibility that miRNAs might be involved in fundamental aspects of motor development, we first searched for miRNAs able to affect the simple locomotor patterns of the *Drosophila* first instar larva. The L1 larva is a convenient model to investigate the genetics of movement given that: (i) the assembly of the machinery for movement must be fully completed by this developmental stage so as to satisfactorily propel the animal into the external world, and (ii) if analysed sufficiently early, for example during the first few minutes after hatching, the animal had no real chance of compensating or learning ways around a putative defect, increasing the probability of detecting a movement phenotype by means of a suitable motor test. To extract a signature of larval movement, we applied a whole animal imaging method based on frustrated total internal reflection (FTIR; *Risse et al., 2017*; *Risse et al., 2013*) which renders high resolution and high contrast movies to both normal and miRNA mutant first instar larvae. This led us to discover that a single miRNA, *miR-2b-1*, had a significant impact on larval movement (*Figure 1B and C*). *miR-2b-1* belongs to the *miR-2* family (*Marco et al., 2012*) and *ΔmiR-2b-1* mutant larvae show a substantial decrease in larval speed (*Figure 1C*) suggesting that absence of this specific genetic component compromises the ability of the larva to move normally. Given that these larval tests were conducted within a 30 min period post-hatching, we considered the possibility that the defects observed in the larvae stemmed from changes in earlier ontogenetic processes that occur in the embryo. More specifically, we decided to test the possibility that early embryonic movement patterns might be affected by the lack of normal expression of *miR-2b-1*.

Previous work had provided an excellent first characterisation of the onset of embryonic movement patterns in wild type embryos (*Crisp et al., 2008*; *Pereanu et al., 2007*); these early studies were based on the manual annotation of representative videotaped muscle contractions (*Pereanu et al., 2007*) or GFP-labelled muscle Z-lines analysed under spinning disc confocal microscopy (*Crisp et al., 2008*). Despite their attributes, these approaches were highly labour intensive and lacked the necessary throughput required to simultaneously analyse large numbers of embryos enabling a sensitive genetic screen. In consequence, we developed a new automated approach capable of quantifying movement in large populations of embryos.

### A high throughput approach to quantify movement in *Drosophila* embryos

To monitor the onset of embryonic movement, which, in normal embryos, occurs during the final third of embryogenesis (i.e. 14–16 hr after egg laying (AEL) *Crisp et al., 2008*; *Pereanu et al.,*

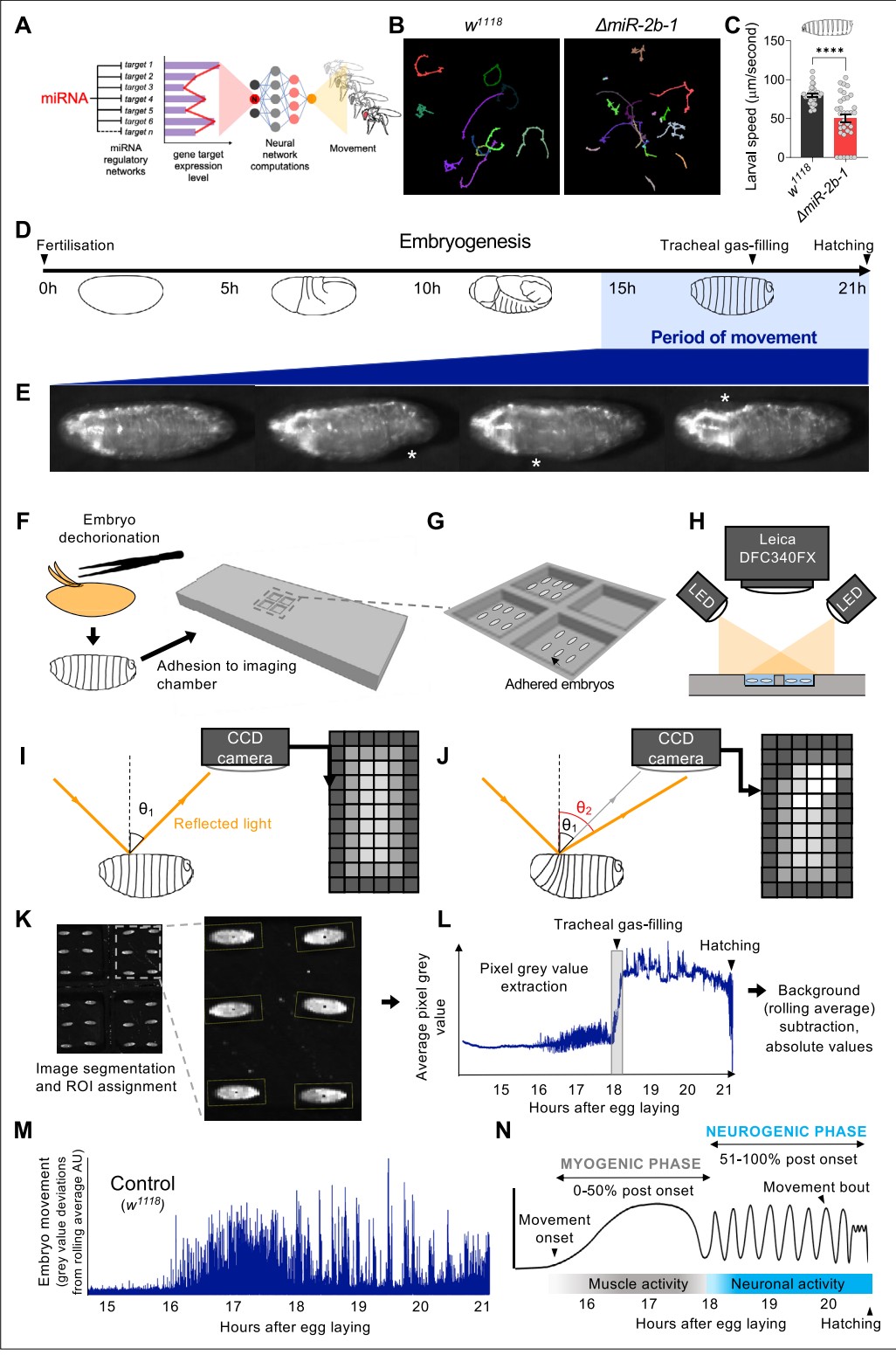

**Figure 1.** A novel approach for the quantification of embryonic movement. (**A**) Diagram illustrating the point of action of miRNAs in the neural networks controlling behaviour. (**B**) Larval movement tracks for *w¹¹¹⁸* (left) and *ΔmiR-2b-1* (right) larvae obtained using the frustrated total internal reflection based imaging system (FIM). (**C**) Quantification of average larval speed for *w¹¹¹⁸* and *ΔmiR-2b-1* larvae using the FIM system. (**D**) Schematic describing the timeline of *Drosophila* embryonic development with the period of movement highlighted in blue. (**E**) Microscope images of a *Drosophila* embryo performing characteristic early movements, highlighted with

*Figure 1 continued on next page*

*Figure 1 continued*

asterisks. (**F–H**) Experimental pipeline for recording embryo movements. (**F**) Eggshell removal (dechorionation) and adhesion to the imaging chamber, (**G**) imaging chamber design and (**H**) imaging set up under incident LED illumination. (**I–J**) Schematic describing the basis for movement detection: light is reflected from the embryo surface and internal structures and detected by a CCD sensor to generate a pixel map of the embryo (**I**). Embryonic movement changes the angle of reflected light, resulting in a different pixel map (**J**). (**K–N**) Pipeline for the quantification of embryonic movement. (**K**) Representative image of the embryo movement chamber showing the assignment of regions of interest (ROIs) to individual embryos; (**L**) extraction of the mean grey value (MGV) for each frame (done in parallel for each ROI) allows the generation of raw movement traces for each individual embryo. Key developmental events that impact MGV (tracheal gas-filling, hatching) are indicated by arrowheads. (**M**) Subtraction of the trace background calculated by rolling average removes slow changes in MGV that result from developmental events, allowing accurate quantification of deviations in MGV from baseline which represent movement over time (absolute values); (**N**) schematic of an idealised wild-type ($w^{1118}$) movement trace with putative phases indicated. The statistical test shown in panel **C** is a Welch's t test, ****=p<0.0001, n=29 to 34.

The online version of this article includes the following source data for figure 1:

**Source data 1.** Larval speed in $w^{1118}$ vs miR-2b-1 mutant genotypes.

*2007*; *Figure 1D and E*) we developed a 3D printed chamber system capable of hosting multiple embryos submerged under a thin layer of halocarbon oil to ensure adequate oxygenation and hydration (*Figure 1F and G*) compatible with digital imaging by a charge coupled device (CCD) camera (*Figure 1H*). The camera captures the reflected light after its physical contact with the embryo; in this setting, even a subtle movement performed by the embryo results in a change in the path of reflected light, leading to variations in signal intensity detected by individual pixels in the CCD sensor, allowing for an accurate measurement of embryonic movement (*Figure 1I and J*). To extract quantitative movement information from individual embryos we applied an image segmentation protocol to define regions of interest (ROIs) corresponding to each embryo and collected pixel intensity values from all ROIs at 4 frames/s (*Figure 1K and L*). The data allow us to plot variations in average grey pixel intensity over embryonic time, which provide a quantitative signature of the ontogeny of movement in the individual embryo (*Figure 1M*). From this, we were able to observe distinct phases of movement that are consistent with previous data: namely, the onset of a phase of disorganised movements ~16 hr after egg laying (hAEL) and its transition into a phase characterised by rhythmic bursts of activity and inactivity ~18 hAEL (*Crisp et al., 2008*; *Pereanu et al., 2007*). These phases have been termed as 'myogenic' and 'neurogenic' based on their respective dependence on neural input (*Figure 1N*; *Crisp et al., 2008*; *Carreira-Rosario et al., 2021*; *Crisp et al., 2011*; *Zeng et al., 2021*), and have been observed in *Drosophila*, as well as in other systems, including vertebrates (*Hamburger, 1963*; *Hamburger and Balaban, 1963*; *Hamburger et al., 1965*; *Ripley and Provine, 1972*; *Saint-Amant and Drapeau, 1998*) strongly suggesting that this is a general feature of motor development. See *Video 1* for a high-resolution recording of *Drosophila* embryo movement.

To establish whether the readings of motor activity at the neurogenic phase detected by our approach were indeed dependent on neural activity, we expressed the inwardly rectifying potassium channel (*Kir*) (*Baines et al., 2001*) in all embryonic neurons using the pan-neuronal driver *elav-Gal4* seeking to suppress action potentials across all embryonic neuronal types. The results of this experiment show that whilst movement patterns during the early chaotic phase remain unchanged by this treatment (*Figure 2C and E*), motor activity at the rhythmic phase is almost completely eliminated (*Figure 2C and F*) strongly indicating that the emergence of this latter phase depends on normal neural activity. This is in agreement with previous observations of the effects of embryonic neural activity inhibition by other methods (*Risse et al., 2013*; *Pereanu et al., 2007*). In addition, spectral analysis demonstrates that the

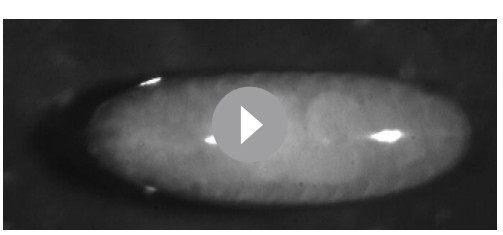

**Video 1.** Patterns of movement during *Drosophila* embryogenesis. Movement from a control $w^{1118}$ embryo recorded across the movement period from 16 to 21 hr after egg laying, 300 X speed.

https://elifesciences.org/articles/95209/figures#video1

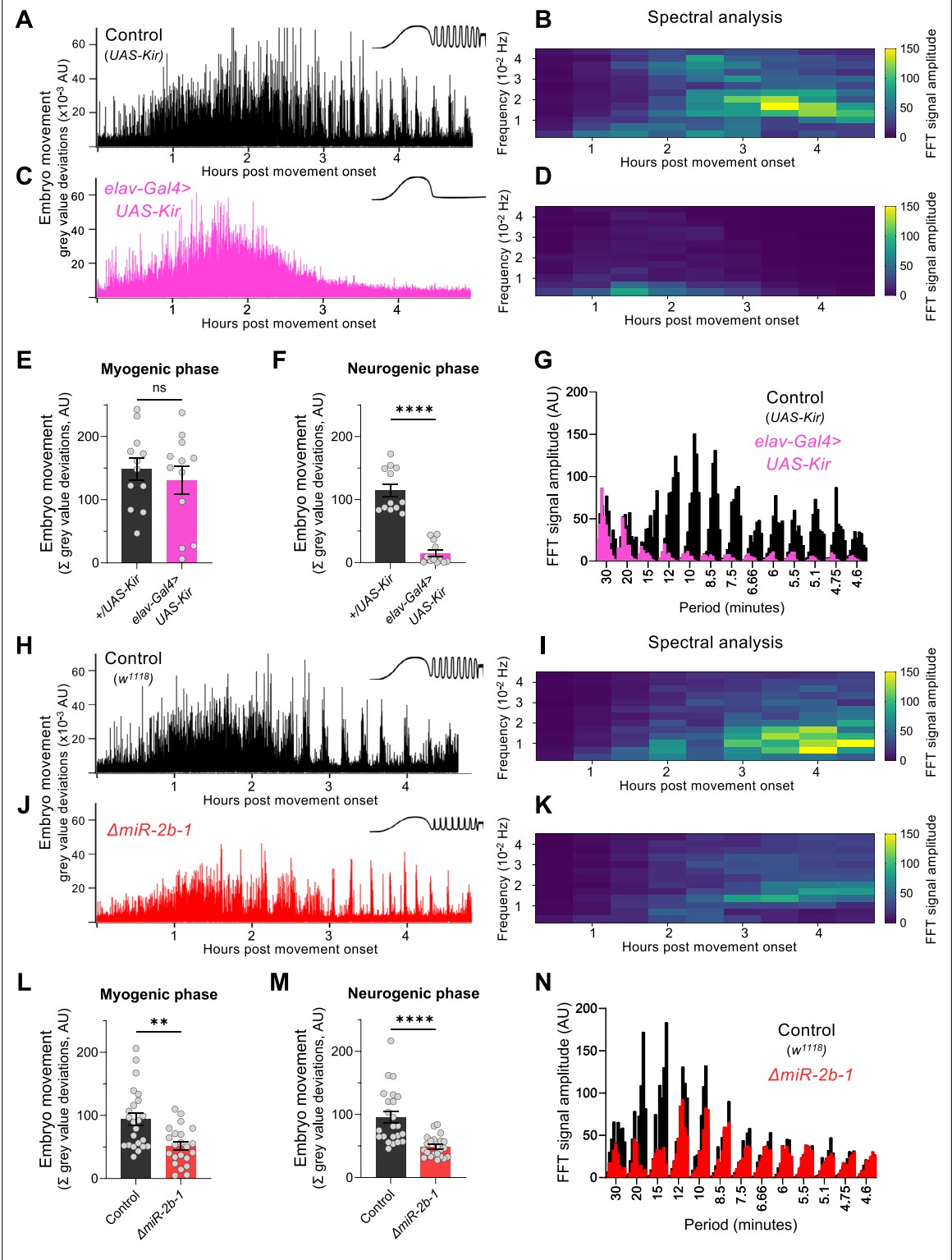

**Figure 2.** miR-2b-1 controls movements during the neurogenic phase of embryonic movement. (**A**) Representative movement trace for control (*UAS-Kir*) animals. A concept diagram that summarises the pattern is shown in the top right. (**B**) Heat map showing the average frequency spectrogram for movements of control (*UAS*-Kir) animals, determined by fast fourier transform (FFT) analysis (1 hr sliding window with a discrete 30 min step size) from onset of embryonic movement to hatching. Brighter colours indicate a stronger amplitude of movement at a given frequency. (**C**) Representative movement trace for experimental (*Elav-Gal4 >UAS* Kir) animals. (**D**) Average frequency spectrogram for *Elav-Gal4 >UAS* Kir animals. (**E**) Summation of MGV deviations during the myogenic phase in control (*UAS-Kir*, black) and experimental (*Elav-Gal4 >UAS* Kir, pink) animals. (**F**) Summation of MGV

*Figure 2 continued on next page*

*Figure 2 continued*

deviations during the neurogenic phase in control (*UAS-Kir*, black) and experimental (*Elav-Gal4 >UAS* Kir, pink) animals. (**G**) Distribution of signal amplitudes across different movement periods (p) derived from the FFT frequency analysis shown in panels **B** and **D**. A higher signal amplitude is produced when more movement occurs with a particular periodicity. Bars of different height at each period sampled show data from individual embryos. (**H**) Representative movement trace for control (*w^1118^*) animals. (**I**) Heat map showing the average frequency spectrogram for movements of *w^1118^* control embryos. (**J**) Representative movement trace for experimental (*ΔmiR-2b-1*) animals. (**K**) Heat map showing the average frequency spectrogram for movements of *ΔmiR-2b-1* embryos. (**L**) Summation of MGV deviations during the myogenic phase in control (*w^1118^*, black) and experimental (*ΔmiR-2b-1*, red) animals. (**M**) Summation of MGV deviations during the neurogenic phase in control (*w^1118^*, black) and experimental (*ΔmiR-2b-1*, red) embryos. (**N**) Distribution of signal amplitudes across different movement periods derived from the FFT frequency analysis shown in panels **I** and **K**. The statistical tests shown in panels **E, F, L** and **M** are multiple Mann-Whitney tests with a Bonferroni-Dunn correction, **=p<0.01, ****=p<0.0001, n=12 in panels **E** and **F**, n=22 to 23 in panels **L** and **M**.

The online version of this article includes the following source data and figure supplement(s) for figure 2:

**Source data 1.** FFT analysis of the +/UAS-Kir genotype.

**Source data 2.** FFT analysis of the elav-Gal4 >UAS Kir genotype.

**Source data 3.** Embryonic movement analysis of +/UAS-Kir vs elav-Gal4 >UAS Kir genotypes.

**Source data 4.** FFT analysis of the w^1118^ genotype.

**Source data 5.** FFT analysis of the miR-2b-1 mutant genotype.

**Source data 6.** Embryonic movement analysis of the w^1118^ vs miR-2b-1 mutant genotypes.

**Figure supplement 1.** miR-2b-1 decreases movement burst duration during the neurogenic phase.

**Figure supplement 1—source data 1.** Embryonic movement average movement burst duration.

movement frequencies that characterise the rhythmic phase (*Figure 2B and G*) do not emerge in *Kir* embryos (*Figure 2D and G*).

## The miRNA *miR-2b-1* affects embryonic movement patterns

Given that the machinery for larval movement is assembled during embryogenesis (*Bate and Martinez Arias, 1993*; *Clark et al., 2018*; *Landgraf et al., 2003*; *Heckscher et al., 2012*), we considered the possibility that *miR-2b-1* might have an impact on the emergence of movement in the fly embryo. To explore this, we applied the approach described above to normal and *ΔmiR-2b-1* mutant embryos (*Figure 2H and J*). These experiments showed that *ΔmiR-2b-1* mutant embryos displayed a different pattern of embryonic movement when compared with their wild type counterparts. Although the distinct phases of embryonic movement are clearly recognisable in mutant embryos, the overall amount of movement appeared greatly reduced (*Figure 2J*).

Indeed, comparison of quantity of movement in wild type and *ΔmiR-2b-1* mutant embryos either during the earlier myogenic phase (*Figure 2L*) or during the neurogenic phase (*Figure 2M*) shows significantly lower levels of movement in mutants. Furthermore, spectral analyses of embryonic movement traces reveal that miRNA mutant embryos shift to a higher frequency of movement bouts during the rhythmic phase when compared to normal embryos (*Figure 2I and K*) and that average bout length is also shortened (*Figure 2—figure supplement 1*). Altogether these observations suggest that miR-2b-1 might exert its roles on embryonic movement, at least in part, due to action within the developing embryonic nervous system.

## *miR-2b-1* expression and roles in the embryonic nervous system

The fact that removal of *miR-2b-1* impacts the neurogenic phase of embryonic movement suggests the possibility that this miRNA might be expressed in the nervous system and exert a functional role there. To further explore this model, we first conducted a spatial expression analysis in the developing embryo using fluorescence RNA in situ hybridisation (FISH). Taking advantage from the fact that *miR-2b-1* is located within the 3' untranslated region (3'UTR) of the *Bruton tyrosine kinase* (*Btk*) gene (*Theroux and Wadsworth, 1992*; *Vincent et al., 1989*; *Figure 3A*), we prepared an in situ probe to detect *Btk* transcripts. These FISH experiments show that the *miR-2b-1* precursor miRNA transcript (pre-miRNA; *Bartel, 2018*) is expressed in multiple embryonic tissues, including the CNS (*Figure 3B*). To further confirm the expression of *miR-2b-1* in the nervous system, we conducted fluorescence-activated cell sorting (FACS, *Figure 3C*) to isolate embryonic neuronal samples (labelled by means of *elav >GFP*) followed by RT-PCR for the mature miRNA transcript and observed *miR-2b-1*-specific

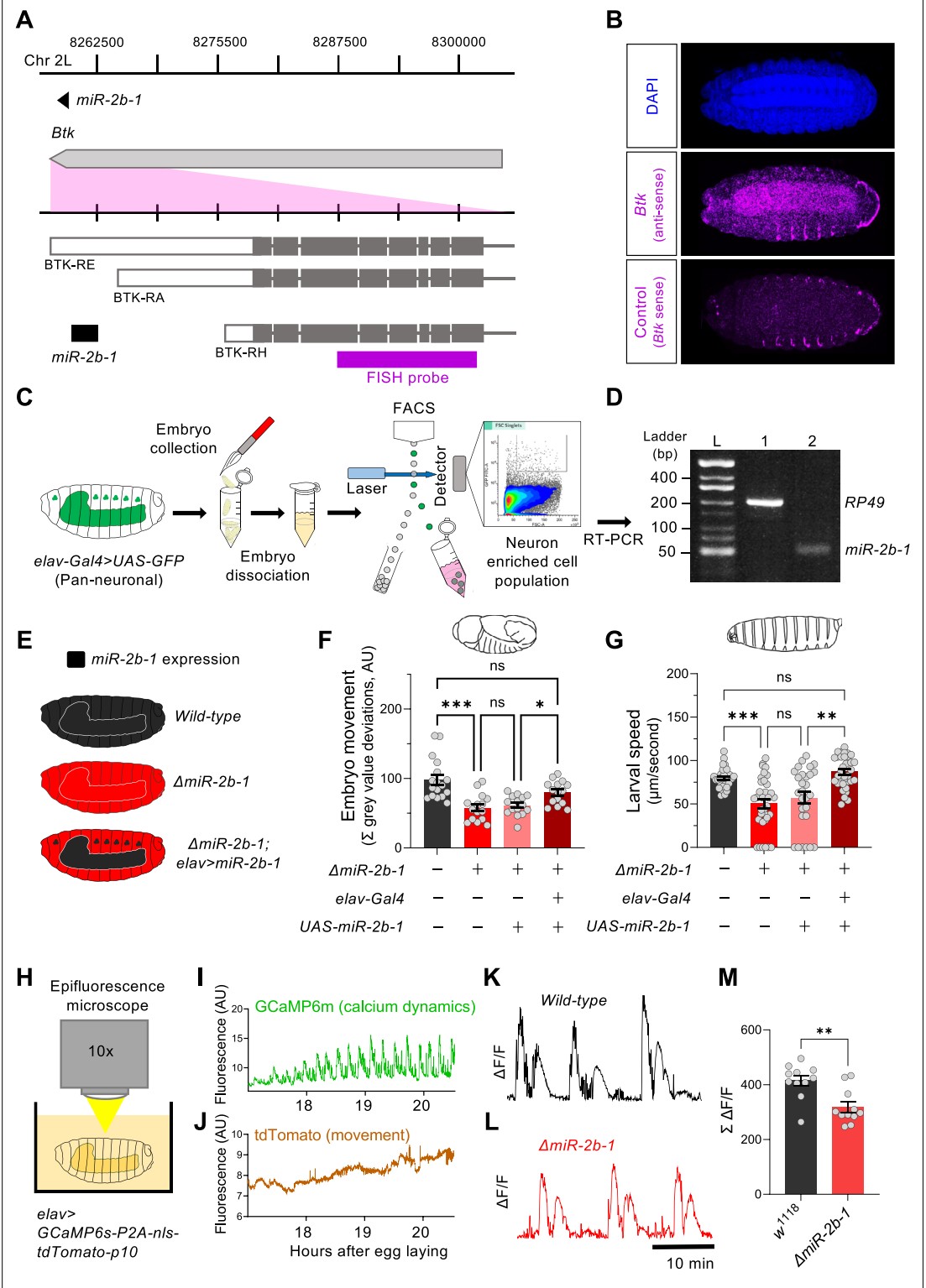

**Figure 3.** miR-2b-1 acts within neurons to regulate embryonic and larval movement. (**A**) Gene diagram describing the *miR-2b-1* locus including the host gene *Btk* and its RNA transcripts. A fluorescence in situ hybridisation (FISH) probe for *Btk* is indicated by the magenta bar. (**B**) FISH experiment on control *w*[1118] embryos showing expression of the miRNA host *Btk* transcripts (antisense probe, middle panel) [DAPI stain in blue (upper image); *Btk* sense probe in magenta (lower image)]. (**C**) Experimental workflow for a fluorescence activated cell sorting (FACS) experiment. Neurons were labelled by *elav-Gal4 >UAS* GFP (left), followed by enzymatic and mechanical separation and isolation of GFP +neurons. (**D**) RT-PCR analysis showing expression of mature *miR-2b-1* (right) detected in neurons at the onset of the neurogenic phase (*RP49* signal (left) shown as control). (**E**) Movement

*Figure 3 continued on next page*

*Figure 3 continued*

patterns were assessed in embryos with normal *miR-2b-1* expression (wild-type, black), null *miR-2b-1* mutants (red) and in mutant embryos in which *miR-2b-1* expression was restored (reconstituted) specifically in neurons (red and black). (**F**) Summation of MGV deviations during the neurogenic phase of embryonic movement in control *w1118* embryos (black bar); Δ*miR-2b-1* mutant embryos (bright red bar); Δ*miR-2b-*, *UAS-miR-2b-1* parental control embryos (faded red bar); Δ*miR-2b-1*, *Elav-Gal4 >UAS-miR-2b-1* embryos (black and red lined bar). (**G**) Average L1 larval speed for the same genotypes used in the embryonic genetic reconstitution experiment. (**H**) Schematic describing the experimental setup for fluorescence imaging of *elav >GCaMP6s-P2A-nls-tdTomato-p10* embryos under an epifluorescence microscope. This design allows simultaneous detection of calcium dynamics (GCaMP6s) and movement (tdTomato). (**I**) Representative GCaMP6s trace (green) from control *w1118* embryos over the neurogenic phase of embryonic movement. (**J**) Representative tdTomato trace from the same embryo as in panel **I**, acting as a passive fluorescence reporter used to subtract changes in GCaMP6s signal induced by embryonic movement. (**K**) Representative ΔF/F trace for *w1118* embryos. (**L**) Representative ΔF/F trace for Δ*miR-2b-1* mutant embryos. (**M**) Summation of ΔF/F signal during embryogenesis in control *w1118* embryos (black bar) and Δ*miR-2b-1* mutant embryos (red bar). The statistical test shown in panel **F** is a Brown-Forsythe and Welch ANOVA with multiple comparisons, *=$p<0.05$, ***=$p<0.001$, n=15 to 17. The statistical test shown in panel **G** is a Kruskal-Wallis test with multiple comparisons, **=$p<0.01$, ***=$p<0.001$, n=28 to 34. The statistical test shown in panel **M** is a Mann-Whitney test, **=$p<0.01$, n=11.

The online version of this article includes the following source data and figure supplement(s) for figure 3:

**Source data 1.** miR-2b-1 transcript (5 p and 3 p respectively, left to right) expression in neurons via end-point PCR.

**Source data 2.** Embryo movement, genetic reconstitution of miR-2b-1 in all neurons.

**Source data 3.** Larval speed, genetic reconstitution of miR-2b-1 in all neurons.

**Source data 4.** Quantified calcium dynamics at embryonic stage.

**Figure supplement 1.** Additional controls for miR-2b-1 genetic reconstitution in neurons; structural analysis in ΔmiR-2b-1 embryos.

**Figure supplement 1—source data 1.** Embryonic movement, parental control for genetic reconstitution of miR-2b-1 in all neurons.

**Figure supplement 1—source data 2.** Larval speed, parental control for genetic reconstitution of miR-2b-1 in all neurons.

**Figure supplement 2.** Inhibiting neural activity during embryogenesis reduces larval locomotor speed.

**Figure supplement 2—source data 1.** Larval speed, no light treatment controls.

**Figure supplement 2—source data 2.** Larval speed, experimental light treatment.

---

signal (*Figure 3D*). Therefore, the results of these two distinct and complementary methods provide strong evidence of neural expression of the miRNA.

To gain more insight on the neural roles of *miR-2b-1* in regard to embryonic movement, we conducted a genetic reconstitution experiment in which we analysed the consequences of restoring *miR-2b-1* expression in the nervous system in an otherwise Δ*miR-2b-1* null mutant (*Figure 3E*). Results in *Figure 3F*, *Figure 3—figure supplement 1A* show that *elav*-driven expression of *miR-2b-1* in a Δ*miR-2b-1* mutant background leads to a phenotypic rescue, producing embryos that display statistically indistinguishable movement patterns to those recorded in control embryos, indicating that neural expression of *miR-2b-1* is sufficient to restore a normal onset of embryonic movement. To further examine the biological roles of neural *miR-2b-1* expression, we assessed the impact of restoring miRNA expression in embryonic neurons on first instar larval locomotor patterns using the FIM approach described above (*Figure 1B and C*) and observed that when Δ*miR-2b-1* mutant larvae are developmentally provided with pan-neuronal *miR-2b-1* expression, the characteristic miRNA larval mutant phenotype is rescued, with specimens moving at natural speed (*Figure 3G*, *Figure 3—figure supplement 1B*). The experiments described above strongly indicate that expression of *miR-2b-1* in the nervous system is biologically relevant and sufficient to rescue the embryonic and larval movement defects observed in Δ*miR-2b-1* mutants. They also suggest that the effects of *miR-2b-1* observed at earlier stages (myogenic phase) are possibly offset by normal neural expression of *miR-2b-1*.

In turn, this suggests that absence of *miR-2b-1* must impinge a morphological and/or a functional deficit in the developing nervous system of the embryo. To tease apart these potential biological effects, we examined the structure of the nervous system in normal and Δ*miR-2b-1* mutant embryos by means of immunohistochemistry and confocal microscopy and observed no detectable differences (*Figure 3—figure supplement 1C*). In contrast, analysis of neural activity patterns in the embryo by means of GCaMP6 functional imaging using a movement distortion correction approach (i.e. tdTomato, *Figure 3H–J*; *Carreira-Rosario et al., 2021*), shows that miRNA mutant embryos have a reduced level of calcium dynamics when compared with their control counter parts (*Figure 3K–M*); notably, this occurs during a previously identified 'critical period' when neural activity levels are of crucial importance to the development of stable neural circuits (*Ackerman et al., 2021*; *Baines and*

*Landgraf, 2021*; *Giachello and Baines, 2015*). Consistently with previous work, artificial reduction of embryonic neural activity via optogenetic control leads to a significant decrease in larval speed (*Figure 3—figure supplement 2A–G*).

## A genetic link between *miR-2b-1* and embryonic movement

Our gene expression, genetic reconstitution, morphological and functional imaging data support a model in which *miR-2b-1* plays a physiological role in the developing embryonic nervous system. This raises the question of how might this regulatory miRNA system interact with the physiological control of the neuron during embryogenesis. To explore the genetic elements that link *miR-2b-1* to its role in embryonic movement, we searched for candidate *miR-2b-1* target genes using the ComiR bioinformatic platform (*Figure 4B*; *Bertolazzi et al., 2020*; *Coronnello and Benos, 2013*). A common issue with bioinformatic predictions of miRNA targets is the generation of false positives; in this regard, ComiR integrates multiple miRNA target prediction algorithms – each one with its intrinsic strengths and weaknesses (*Betel et al., 2010*; *Kertesz et al., 2007*; *McGeary et al., 2019*; *Miranda et al., 2006*) – seeking to identify a set of consistent *bona fide* miRNA targets that satisfy the filters of multiple algorithms, thus reducing the generation of false positives (*Bertolazzi et al., 2020*; *Coronnello and Benos, 2013*). Applying ComiR to *miR-2b-1* produced a list of high probability targets organised in the form of an ascending ranking (*Figure 4B*). At the very top of the list was *CG3638*, an uncharacterised *Drosophila* gene predicted to encode a chloride channel protein; this highly ranked target was of interest to us because of its potential role in the physiological control of anionic conductances, and its broad evolutionary conservation across insects and mammals (*Suzuki, 2006*), including humans (*Figure 4E–I*).

A genuine genetic target for a given miRNA is predicted to: (i) be de-repressed (up-regulated) in the absence of the miRNA; and (ii) be down-regulated under miRNA ectopic expression. Analysis of *CG3638* expression shows that this target meets the predictions of a genuine *miR-2b-1* target in full: expression of *CG3638* is upregulated in *ΔmiR-2b-1* mutants (*Figure 4B*) and reduced under neural over-expression of *miR-2b-1* (*Figure 4D*). As mentioned above, phylogenetic analysis of *CG3638* reveals that it belongs to an evolutionarily conserved family of chloride channel genes (*Suzuki, 2006*; *Suzuki and Mizuno, 2004*), with representatives across distantly related lineages of insects and vertebrates including mammals, strongly indicating a functional role (*Figure 4E*). Comparison of the properties of *CG3638* and its human orthologue reveal the characteristic seven trans-membrane domains with an external carboxyterminal topology (*Figure 4F and H*) further supporting orthology, and applying *AlphaFold* – an artificial intelligence computational method able to predict protein structures with atomic accuracy (*Jumper et al., 2021*) – to the proteins encoded by the *Drosophila CG3638* gene and the human TTYH1 gene reveals the marked similarities between these two polypeptides (*Figure 4G and I*).

To explore the relationship between *CG3638* and the embryonic movement phenotype displayed by *ΔmiR-2b-1* mutants, we tested the effects of an artificial reduction of *CG3638* in the genetic background of the miRNA mutant (*Figure 4K*). In this scenario, should the levels of expression of *CG3638* be relevant to the triggering of the embryonic movement phenotype, we predicted that a reduction in *CG3638* expression levels should compensate its cellular effects, and, accordingly, reduce or even rescue the embryonic phenotype. The results of this experiment show that this is indeed the case, with embryonic movement of the *ΔmiR-2b-1* mutant effectively rescued by a reduction in *CG3638* (*Figure 4J*). Based on its modulatory role in embryonic movement we termed *CG3638* as <u>Mo</u>vement modula<u>tor</u> (*Motor*).

## Mapping the focus of action of *miR-2b-1* within the known networks underlying embryonic movement

Having observed the effects of *miR-2b-1* on embryonic movement we wondered about the site of action of this miRNA in regard to circuit components previously linked to embryonic movement. In this respect, previous work has identified embryonic motor neuronal components, as well as interneurons and elements of the sensory system as playing key roles in the control of motor development. These include a motor component that includes all embryonic motor neurons which command the stereotypic array of muscles in the embryonic body wall (*Landgraf and Thor, 2006*; *Peron et al., 2009*), as well as specific elements of the sensory system – in particular the chordotonal system – which detect

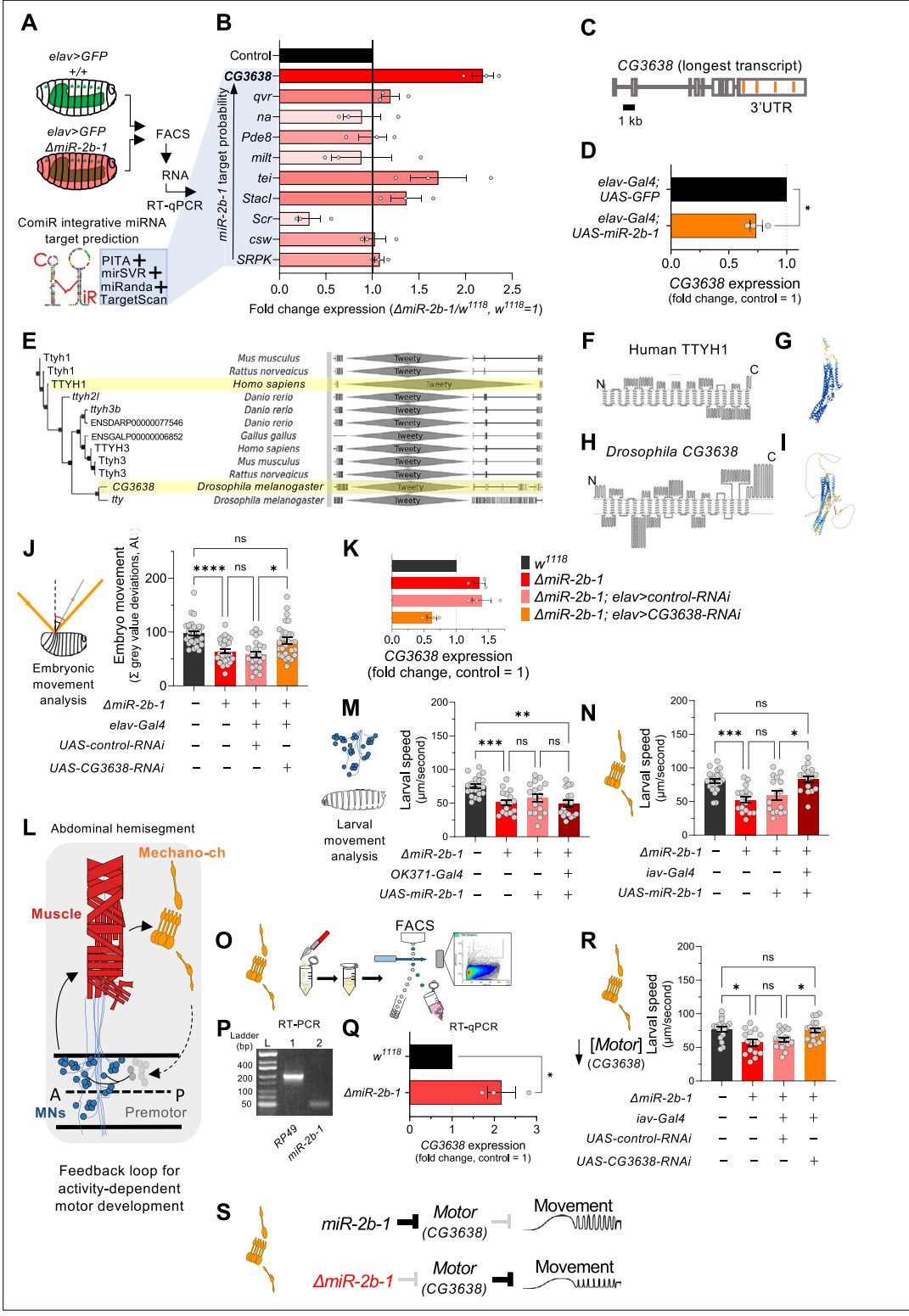

**Figure 4.** The genetic and cellular mechanisms that link miR-2b-1 to embryonic movement. (**A**) Workflow for the FACS and RT-qPCR experiments shown in panel **B** and schematic describing the ComiR miRNA target prediction tool used to generate the list of candidate *miR-2b-1* targets. (**B**) Expression analysis (qPCR) of 10 predicted *miR-2b-1* target genes shown as fold change between *ΔmiR-2b-1* mutant and control *w^{1118}* embryos (three biological replicates). Targets are listed from top to bottom by descending probability score. [The black control bar, set to 1, represents expression of each gene in control *w^{1118}* embryos]. Note that upregulation of *CG3638* is statistically

*Figure 4 continued on next page*

*Figure 4 continued*

significant (p=0.0169). (**C**) Schematic of the *CG3638* transcript with *miR-2b-1* target sites indicated (orange lines). (**D**) Whole embryo qPCR experiment showing a reduction of *CG3638* expression in *elav-Gal4 >UAS-miR-2b-1* embryos (orange bar), relative to control *elav-Gal4 >UAS* GFP embryos (black bar). (**E**) Evolutionary conservation of the *CG3638* protein across a wide range of invertebrate and vertebrate species (left), as determined with PhylomeDB 5 software (**Huerta-Cepas et al., 2014**) [*Homo sapiens* and *Drosophila melanogaster* highlighted in yellow]. Gene schematics highlighting the conserved Tweety domain are shown on the right. (**F–I**) Transmembrane domain structure (left) and AlphaFold structural predictions (right) for Human TTYH1 (**F–G**) and *Drosophila CG3638* (**H–I**). (**J**) Embryonic movement quantification (summation of MGV deviations) of *ΔmiR-2b-1, elav >CG3638* RNAi embryos (orange bar) during the neurogenic phase compared to control *w1118* (black bar), *ΔmiR-2b-1* mutant (bright red bar) and control *ΔmiR-2b-1, elav >control* RNAi embryos (faded red bar). (**K**) qPCR expression profiling of *CG3638* in whole embryos of the genotypes tested in panel **J**. (**L**) Diagram describing key cell types that form a feedback loop for activity-dependent motor development. Motor neurons (MNs, blue) induce muscle (red) movements which are in turn detected by proprioceptive chordotonal organs (Mechano-ch, orange) and feed-back into the CNS to regulate activity patterns. (**M–N**) Reconstitution experiments that restore *miR-2b-1* expression in specific cellular elements related to embryonic movement circuitry. (**M**) Quantification of larval speed in control *w1118* (black); *ΔmiR-2b-1* mutant (red); *ΔmiR-2b-1, UAS-miR-2b-1* parental control (pink) and *ΔmiR-2b-1, OK371-Gal4>UAS-miR-2b-1* experimental embryos (brown). (**N**) Quantification of larval speed in *ΔmiR-2b-1, iav-Gal4 >UAS-miR-2b-1* (brown) and control genotypes as in panel **M**. (**O**) Schematic describing FACS isolation of embryonic chordotonal organs. (**P**) Mature *miR-2b-1* (right) is expressed in chordotonal organs isolated during the neurogenic phase (RP49 expression shown on left). (**Q**) Chordotonal specific qPCR expression profiling of *CG3638* in *ΔmiR-2b-1* mutant and control *w1118* embryos. (**R**) Average larval speed of *ΔmiR-2b-1, elav >CG3638* RNAi (orange) compared to control *w1118* (black), *ΔmiR-2b-1* mutant (red) and control *ΔmiR-2b-1, UAS-control-RNAi* (pink). (**S**) Model for the mechanism by which *miR-2b-1* acts to control embryonic movement in chordotonal organs. Under normal (control) conditions (top), *miR-2b-1* inhibits the expression of *CG3638* and thereby enables normal movement. In *ΔmiR-2b-1* mutants (bottom), de-repression of *CG3638* expression leads to a reduction in embryonic movement. The statistical tests shown in panels **D** and **Q** are Welch's t tests, *=p<0.05, n>10 embryos per biological replicate. The statistical tests shown in panels **J, M, N** and **R** are Brown-Forsythe and Welch ANOVA tests with multiple comparisons, *=p<0.05, **=p<0.01, ***=p<0.001, ****=p<0.0001, n=16 to 28.

The online version of this article includes the following source data and figure supplement(s) for figure 4:

**Source data 1.** Embryo movement, RNAi knockdown of CG3638 in all neurons.

**Source data 2.** Larval speed, genetic reconstitution of miR-2b-1 in motor neurons.

**Source data 3.** Larval speed, genetic reconstitution of miR-2b-1 in chordotonal organs.

**Source data 4.** miR-2b-1 transcript (5 p and 3 p respectively, left to right) expression in chordotonal organs via end-point PCR.

**Source data 5.** Larval speed, RNAi knock down of CG3638 in chordotonal organs.

**Figure supplement 1.** Additional controls for genetic reconstitutions in motor neurons and chordotonal organs.

**Figure supplement 1—source data 1.** Larval speed, parental control for genetic reconstitution of miR-2b-1 in motor neurons.

**Figure supplement 1—source data 2.** Larval speed, parental control for genetic reconstitution of miR-2b-1 in chordotonal organs.

early myogenic movements in the embryo and transmit the information to the pattern generators thus modulating motor patterns (*Figure 4L*; *Carreira-Rosario et al., 2021*; *Zeng et al., 2021*; *Caldwell et al., 2003*; *Hughes and Thomas, 2007*). To determine which one of these known circuitry elements might be the principal focus of action of *miR-2b-1* in connection to embryonic movement control we artificially expressed *miR-2b-1* in each motor and sensory circuit elements – using drivers *OK371-Gal4* (*Mahr and Aberle, 2006*) and *iav-Gal4* (*Kwon et al., 2010*), respectively – in an otherwise null mutant background for *miR-2b-1*, asking whether these genetic restorations were sufficient to improve or perhaps even rescue normal movement patterns. To ensure that circuit-specific Gal-4 drivers were active and UAS-driven miRNA levels achieved the necessary cumulative values for biological activity, we chose to measure effects on early larval movement patterns tested in 30-min-old first instar larvae (L1s). The results of these experiments are shown in *Figure 4M and N*, *Figure 4—figure supplement 1A and B*. Here we observe that restoring expression of *miR-2b-1* in the motor neuronal domain defined by the OK371 driver is insufficient to affect the defective movement patterns observed in *miR-2b-1* null mutants (*Figure 1B and C* and *Figure 4M*, *Figure 4—figure supplement*

1A). In contrast, re-establishing expression of the miRNA in all eight chordotonal organs leads to a full rescue of the motor phenotype (*Figure 4N*, *Figure 4—figure supplement 1B*) suggesting that this aspect of the sensory system might be the one where *miR-2b-1* exerts relevant actions during the normal development of movement. In line with this, we observe that the mature *miR-2b-1* transcript is indeed expressed in FACS-isolated embryonic chordotonal organs prepared from wild type embryos (*Figure 4P*) and, that the genetic target of *miR-2b-1*, *Motor*, is also expressed in these cells in normal embryos (*Figure 4Q*, top). In addition, expression of *Motor* in chordotonal organs prepared from *miR-2b-1* null mutants is up-regulated (*Figure 4Q*, bottom) lending further support to a model in which *Motor* is de-repressed in these specific sensory elements. Furthermore, artificial reduction of *Motor* implemented as a stratagem to decrease the effects of de-repression specifically within the chordotonal system is sufficient to rescue normal movement patterns (*Figure 4R*). Altogether, these findings strongly suggest that *miR-2b-1* impacts the emergence of embryonic movement, at least in part, via effects on the sensory circuit components that underlie motor development, rather than affecting the actual generation of motor patterns.

Our work identifies a miRNA system that plays a role in the emergence of embryonic movement in the fly embryo, and offers a new approach to analyse the roles of non-coding RNAs and protein coding genes at the critical period when patterned movement develops. It has not escaped our attention that our platform may be suitable for testing the effects of drugs and compounds on early motor activity and are indeed exploring this possibility. We are also using the embyonic pipeline reported here to characterise the motor roles of all miRNAs expressed in the Drosophila embryo, seeking to determine the general rules of miRNA action on the emergence of embryonic movement. In complementary work we are also establishing the effects of all Drosophila miRNAs in the movement of young L1 larvae aiming at relating embryonic and larval effects of individual miRNAs. Understanding the molecular elements controlling the onset of motor development in *Drosophila* will put us one step closer to understanding the molecular basis of embryonic movement in other species, including vertebrates, whose embryos seem to undergo remarkably similar transition phases to those reported here (*Hamburger, 1963*).

## Materials and methods

### Key resources table

| Reagent type (species) or resource | Designation | Source or reference | Identifiers | Additional information |
|---|---|---|---|---|
| Genetic reagent (*D. melanogaster*) | w[1118] | Bloomington *Drosophila* Stock Center | BDSC:5905 | Flybase ID: FBst0005905 |
| Genetic reagent (*D. melanogaster*) | w[1118]; TI{w[+mW.hs]=TI} mir-2b-1[KO] | Bloomington *Drosophila* Stock Center | BDSC:58915 | Flybase ID: FBst0058915 |
| Genetic reagent (*D. melanogaster*) | w[1118]; P{w[+mC]=GAL4 elav.L}3 | Bloomington *Drosophila* Stock Center | BDSC:458 | Flybase ID: FBst0000458 |
| Genetic reagent (*D. melanogaster*) | w[1118]; P{w[+mW.hs]=GawB}VGlut[OK371] | Bloomington *Drosophila* Stock Center | BDSC:26160 | Flybase ID: FBst0026160 |
| Genetic reagent (*D. melanogaster*) | w[*]; P{w[+mC]=iav-GAL4.K}3 | Bloomington *Drosophila* Stock Center | BDSC:52273 | Flybase ID: FBst0052273 |
| Genetic reagent (*D. melanogaster*) | UAS-Kir | Bate Lab, Cambridge *Baines et al., 2001* | N/A | |
| Genetic reagent (*D. melanogaster*) | w[1118]; P{y[+t7.7] w[+mC]=10XUAS-IVS-myr::GFP}attP2 | Bloomington *Drosophila* Stock Center | BDSC:32197 | Flybase ID: FBst0032197 |

*Continued*

| Reagent type (species) or resource | Designation | Source or reference | Identifiers | Additional information |
|---|---|---|---|---|
| Genetic reagent (*D. melanogaster*) | w[1118]; P{y[+t7.7] w[+mC]=UAS-LUC-mir-2b-1.T}attP2 | Bloomington *Drosophila* Stock Center | BDSC:41128 | Flybase ID: FBst0041128 |
| Genetic reagent (*D. melanogaster*) | w[1118]; UAS-IVS-Syn21-GCaMP6s-P2A-nls-tdTomato-p10 (JK66B) | Zlatic Lab, Cambridge | N/A | |
| Genetic reagent (*D. melanogaster*) | UAS 40D RNAi-KK | Vienna *Drosophila* Resource Centre | VDRC: 60101 | Flybase ID: FBst0060101 |
| Genetic reagent (*D. melanogaster*) | CG3638 RNAi-KK | Vienna *Drosophila* Resource Centre | VDRC: 102444 | Flybase ID: FBst0474313 |
| Genetic reagent (*D. melanogaster*) | UAS-GtACR2 | Bloomington *Drosophila* Stock Center | BDSC:92984 | Flybase ID: FBst0092984 |
| Antibody | anti-Elav (mouse monoclonal) | Developmental Studies Hybridoma Bank | DSHB: 9F8A9 | IF(1:100) |
| Antibody | anti-Fasciclin II (mouse monoclonal) | Developmental Studies Hybridoma Bank | DSHB: 1D4 | IF(1:100) |
| Antibody | anti-BP102 (mouse monoclonal) | Developmental Studies Hybridoma Bank | DSHB: BP102 | IF(1:100) |
| Antibody | anti-DIG-POD (Fab fragments from sheep) | Roche | 11207733910 | IF(1:500) |
| Antibody | anti-mouse Alexa Fluor 488 (goat) | Invitrogen | A-11001 | IF(1:1000) |
| Antibody | anti-mouse Alexa Fluor 555 (goat) | Invitrogen | A-21426 | IF(1:1000) |
| Sequence-based reagent | *Btk* | Sigma-Aldrich | In-situ hybridisation control sense probe F | ATTTAGGTGACACTATAGAG AATTCAACGCGCAGCATC |
| Sequence-based reagent | *Btk* | Sigma-Aldrich | In-situ hybridisation control sense probe R | ACACCAAACTGTCCCGATCC |
| Sequence-based reagent | *Btk* | Sigma-Aldrich | In-situ hybridisation experimental anti-sense probe F | AGAATTCAACGCGCAGCATC |
| Sequence-based reagent | *Btk* | Sigma-Aldrich | In-situ hybridisation experimental anti-sense probe R | ATTTAGGTGACACTATAGACACC AAACTGTCCCGATCC |
| Sequence-based reagent | Reverse transcription primer | Sigma-Aldrich | miRNA PCR RT mix primer 1 | CAGGTCCAGTTTTTTTTTTTTTTT VN, where V is A, C and G and N is A, C, G and T. |
| Sequence-based reagent | RP49 (RpL32) | Sigma-Aldrich | PCR primer F | CCAGTCGGATCGATATGCTAA |
| Sequence-based reagent | RP49 (RpL32) | Sigma-Aldrich | PCR primer R | TCTGCATGAGCAGGACCTC |
| Sequence-based reagent | miR-2b-1–5 p | Sigma-Aldrich | PCR primer F | GGTCTTCAAAGTGGCAGTG |

*Continued on next page*

*Continued*

| Reagent type (species) or resource | Designation | Source or reference | Identifiers | Additional information |
|---|---|---|---|---|
| Sequence-based reagent | *miR-2b-1–5* p | Sigma-Aldrich | PCR primer R | GTCCAGTTTTTTTTTTTTTTTCATGTC |
| Sequence-based reagent | *CG3638* | Sigma-Aldrich; FlyPrimerBank | PCR primer F; FPB: PP20655 | TCCTTGGTCATCATTACGCTGA |
| Sequence-based reagent | *CG3638* | Sigma-Aldrich; FlyPrimerBank | PCR primer R; FPB: PP20655 | CCATTATGGAAATCATCGTTGCC |
| Sequence-based reagent | *qvr* | Sigma-Aldrich; FlyPrimerBank | PCR primer F; FPB: PP25844 | CCTTTCAACTATACAGCCCTGC |
| Sequence-based reagent | *qvr* | Sigma-Aldrich; FlyPrimerBank | PCR primer R; FPB: PP25844 | TGTAACTGTGACGTACACATGC |
| Sequence-based reagent | *na* | Sigma-Aldrich; FlyPrimerBank | PCR primer F; FPB: PP34188 | ACCTTTCCTCGCGGATTACG |
| Sequence-based reagent | *na* | Sigma-Aldrich; FlyPrimerBank | PCR primer R; FPB: PP34188 | CCACAGCTTGTTCACCCAC |
| Sequence-based reagent | *Pde8* | Sigma-Aldrich; FlyPrimerBank | PCR primer F; FPB: PP11165 | CCGAGAAAATCCGTCCAGC |
| Sequence-based reagent | *Pde8* | Sigma-Aldrich; FlyPrimerBank | PCR primer R; FPB: PP11165 | CAGCGGTCTTGGTCTTTCATTA |
| Sequence-based reagent | *milt* | Sigma-Aldrich; FlyPrimerBank | PCR primer F; FPB: PP21284 | GCAGACGATGGCACAGATACT |
| Sequence-based reagent | *milt* | Sigma-Aldrich; FlyPrimerBank | PCR primer R; FPB: PP21284 | CGTCGAGCAGGGAGTTGAC |
| Sequence-based reagent | *CG17716* | Sigma-Aldrich; FlyPrimerBank | PCR primer F; FPB: PP26416 | GTCCGTGGTCTATGCGGAG |
| Sequence-based reagent | *CG17716* | Sigma-Aldrich; FlyPrimerBank | PCR primer R; FPB: PP26416 | ATGAAGCGATAGTCGGTGACG |
| Sequence-based reagent | *Stacl* | Sigma-Aldrich; FlyPrimerBank | PCR primer F; FPB: PP10900 | GCTGCGTCCCAATCTGGAT |
| Sequence-based reagent | *Stacl* | Sigma-Aldrich; FlyPrimerBank | PCR primer R; FPB: PP10900 | CGTGTGTGCCCTCTCAGAAT |
| Sequence-based reagent | *Scr* | Sigma-Aldrich; FlyPrimerBank | PCR primer F; FPB: PP19886 | GGCGGCCTATACGCCTAAC |
| Sequence-based reagent | *Scr* | Sigma-Aldrich; FlyPrimerBank | PCR primer R; FPB: PP19886 | CGGCTGTAGCTGCGTGTAG |
| Sequence-based reagent | *csw* | Sigma-Aldrich; FlyPrimerBank | PCR primer F; FPB: PP8739 | TTTGGCACCTTGTCGGAACT |
| Sequence-based reagent | *csw* | Sigma-Aldrich; FlyPrimerBank | PCR primer R; FPB: PP8739 | CCAGAAACCTCCCTTGACCAG |
| Sequence-based reagent | *SRPK* | Sigma-Aldrich; FlyPrimerBank | PCR primer F; FPB: PA60244 | ATCCGCTGACTGAGGGCACTG |
| Sequence-based reagent | *SRPK* | Sigma-Aldrich; FlyPrimerBank | PCR primer R; FPB: PA60244 | GTAGAGTTTTCCAGTTGTGG |

## Experimental model details

*Drosophila melanogaster* fruit flies were maintained by standard means; in 25 °C incubators with 50–60% humidity; on a 12 hr light/dark cycle; with molasses food. See reagent and resource table for all *Drosophila* strains used in this project and the respective sources.

## Collection of samples for behavioural experiments

Flies were kept at 25 °C in collection cages with food plates consisting of apple juice agar and yeast paste. Embryos were collected by placing a fresh food plate in the collection cage and allowing flies to lay eggs for 1 hr. Prior to all embryo collections, a pre-collection of 1 hr was performed to reduce female egg storage. In experiments where some embryos were of genotypes that included GFP-tagged balancer chromosomes, those individuals were selected against by fluorescence microscopy. Selected embryos were gently moved to a fresh plate and allowed to develop at 25 °C. All genotypes underwent selection by fluorescence microscopy to ensure consistent exposure to light across groups compared.

## Embryo chamber design and 3D-printing

The 3D-printed embryo chamber was designed on paper to dimensions of 45 mm (L) X 15 mm (W) X 3 mm (D). Four sub-chambers were designed within the main chamber, each with dimensions of 5 mm (L) X 5 mm (W) X 0.5 mm (D) and divided by a boundary wall of 0.4 mm. The design was subsequently coded in OpenSCAD software and printed on a Formlabs Form 2 desktop 3D-printer using biocompatible BioMed Black resin.

## Embryo movement experiments

Embryo collections were aged to 14 (±0.5) hours after egg laying (AEL) prior to selection of individuals with the correct genotype determined via the fluorescence balancer. Embryos were subsequently adhered to a piece of tesa double-sided tape that itself was adhered to a microscope slide. Using one end of a pair of dissection forceps and observing through a brightfield microscope, embryos were gently rolled on the tape to break them out of the egg chorion before being transferred to a well of an embryo chamber previously glued with tesa glue dissolved in heptane. 6 embryos were transferred one-by-one to the well prior to the addition of 3 µl of Halocarbon oil (a 50:50 mix of series 27 and series 700). All manipulations, from the dechorionation of the first embryo to the addition of Halocarbon oil, were done within 3 min to ensure minimal dehydration of embryos. This process was repeated for each of the remaining chamber wells.

## Embryo movement recording

Movements of embryos across all four wells of the embryo chamber were recorded simultaneously using a Leica DFC 340 FX camera mounted on a Leica M165 FC microscope, with a resolution of 480x360 pixels and a frame rate of 4 frames per second. Incident lighting was directed laterally onto the embryo chamber to avoid glare to the camera from the surface of the Halocarbon oil. Consistent lighting conditions across the four wells of the embryo chamber were ensured through measurement of pixel intensities (mean grey values) within each recorded well in ImageJ software. Recordings were carried out for at least 10 hr to capture the entire duration of embryonic movement up until larval hatching and files were stored in the AVI format with MJPEG compression to ensure compatibility with downstream analysis software. All recordings were carried out at 25 °C.

## Embryo movement analysis

AVI files were opened in ImageJ software and a rectangular ROI of consistent size was applied over each embryo within the chamber. The 'RoiSet' of up to 24 ROIs was saved and then used to 'multi-measure' the mean grey value (MGV) of each ROI for each frame of the recording – rapid changes within which were caused by embryo movements that altered reflected light to the camera. The resulting list of MGV was exported to Excel software where a background subtraction was applied to remove slow-scale changes to MGV that occurred due to gradual changes in embryo morphology. This involved the generation of a moving average for the MGV of each embryo with a sliding-window of 60 frames or 15 s, which was then subtracted from the MGV for each frame. The choice of 15 s was made empirically based on the duration of individual movements and the rate of morphological change, particularly tracheal gas-filling. Absolute values were taken for deviations from the baseline to create traces that represent embryo movements over time and for all quantifications. Traces were subsequently cropped at larval hatching based on when the vitelline membrane was breached by the head of the larva. Movement was quantified by summing deviations in MGV from baseline prior to larval hatching. A threshold value of 0.01 MGV deviations from baseline, determined empirically by

the comparison of traces from unfertilised and live embryos and found to be applicable across recordings, was applied to filter noise that was unrelated to embryo movement. Traces where a different larva had entered the ROI following hatching were removed from the analysis. Fast Fourier Transform (FFT) analysis was performed in Igor PRO software and was applied to 1 hr overlapping (30 min overlaps) sliding windows of movement traces to extract information about the frequency spectrum of movements.

## Larval movement experiments

For all larval movement experiments, we used an imaging method based on frustrated total internal reflection (FTIR) – known as FIM (FTIR-based Imaging Method). This allowed for the quantification of larval movement with a high degree of consistency and accuracy. (See references *Risse et al., 2017*; *Risse et al., 2013* for more information). A FIM table was obtained from the University of Münster, department of Computer Vision and Machine Learning Systems, for this purpose. First instar larvae were gently moved to fresh agar plates for assessment on the FIM table within 30 min of hatching to ensure consistency of age across larvae tested and reduce the possibility of differences in motor learning. At least 25 larvae were assessed for each genotype across 3 independent recordings. TIFF images were captured for 3 min at a resolution of 1200x1200 pixels and frame rate of 7 frames per second using a Basler acA2040-90um camera. All recordings were carried out at 25 °C in low-light conditions.

## Larval movement analysis

TIFFs were opened in FIMTrack software (*Risse et al., 2017*) before running the tracking algorithm with the 'minimum larval size' set to 40. All other settings were left as default. Partial tracks, due to larvae crawling off the plate or into one another, were removed from the analysis. Larvae that did not move were considered a 0 value as the FIMTrack software was unable identify them. The L1_acc_dis parameter was extracted for each larva and this was taken as a quantitative measure of larval movement and compared across genotypes.

## Statistical analyses

All statistical analyses were performed in GraphPad Prism software. The normality of each dataset was determined by the agreement of four tests: D'Agostino & Pearson; Anderson-Darling; Shapiro-Wilk; Kolmogorov-Smirnov. Datasets that at least one of these normality tests identified as not having a normal distribution were further assessed by nonparametric tests. Multiple Mann-Whitney tests with Bonferroni correction were used for comparison of two genotypes in the myogenic and neurogenic phases or at the larval stage. The parametric Brown-Forsythe and Welch ANOVA with Dunnett's T3 multiple comparisons tests, or nonparametric Kruskal-Wallis ANOVA with Dunn's multiple comparisons tests were used for comparisons of more than two genotypes assessed in parallel during the miRNA rescue and RNAi experiments. ****=$p < 0.0001$, ***=$p < 0.001$, **=$p < 0.01$, *=$p < 0.05$.

## In-situ hybridisation

In situ hybridisation probes were designed to be 500–1000 bases in length and complementary to the exons of *Btk* mRNA. A negative control probe made in the sense orientation to the target mRNA was used alongside the experimental anti-sense probe, at the same concentration, to control for non-specific binding. See reagent and resource table for all primers used in probe synthesis. An SP6 polymerase tag was added to the forward (sense probe) or reverse (anti-sense probe) primer for transcription. $w^{1118}$ embryos were pre-hybridised in hybridisation solution (50% formamide) for at least 2 hr prior to overnight hybridisation at 55 °C. Post-hybridisation, embryos were blocked and stained with an α-DIG-POD antibody (Roche 11207733910), prior to a fluorescein tyramide treatment to increase signal strength and imaging with a confocal fluorescence microscope.

## Fluorescence activated cell sorting (FACS)

For cell dissociation, embryos were collected and aged to 18 (±0.5) hours AEL prior to dechorionation and digestion in a haemolymph-like solution (90 mM NaCl, 25 mM KCl, 10 mM HEPES, 80 mM D-glucose, 4.8 mM NaHCO3, pH 7) with 0.25% trypsin at 37 °C and gentle mechanical disruption. The cell solution was passed through a 40 nm filter immediately prior to sorting. Cell sorting was performed on

a BD FACSmelody cell sorter (BD Biosciences) calibrated to sort GFP + cells by sorting 100,000 cells from embryos of the *UAS-GFP* genotype (without a Gal4 driver) and observing the highest level of fluorescence seen from these cells, before gating the cell sorter to only isolate cells with a level of fluorescence above this. For each biological replicate of each genotype, 10,000 cells were sorted into 470 µl TRIzol reagent (Invitrogen) for downstream RNA extraction.

## Conventional and real-time quantitative PCR

Conventional PCR was performed with standard Taq DNA polymerase (New England Biolabs – M0273) For all reactions, 30 amplification cycles were run with 0.4 µM final concentration of each primer (see reagent and resource table for a list of all primers used) and a 60 °C annealing temperature. qPCR reactions were performed with LightCycler SYBR Green I reagents (Roche – 04707516001). For all reactions, 40 amplification cycles were run with 0.25 µM final concentration of each primer (see reagent and resource table for all primers used) and a 60 °C annealing temperature. All reactions were run with 2 technical replicates and any groups compared in downstream analysis were run on the same reaction plate. Continuous melt curves were examined to assess whether a single amplicon was amplified by each primer set and no-template controls were also run to confirm a lack of primer-dimer formation. Primer efficiency was determined by a standard curve of 6 cDNA dilutions and only those with efficiencies between 1.9 and 2.2 were used. Efficiency – $E$ – was calculated with the following equation (*Pfaffl, 2001*):

$$E = d^{-1/-s}$$

where $d$ is the dilution factor and $s$ is the slope of the curve. Fold change in transcript expression between two experimental conditions was calculated using $C_T$ values obtained from the qPCR experiment with the following equation (*Pfaffl, 2001*):

$$Fold\ change = \frac{2^{CT\ gene\ of\ interest\ (control-mutant)}}{2^{CT\ reference\ gene\ (control-mutant)}}$$

## Mature miRNA PCR

PCR to specifically amplify mature miRNA transcripts utilised a protocol based on a single reverse transcription reaction for all microRNAs combined with PCR using two, mature miRNA-specific DNA primers (*Balcells et al., 2011*). Poly(A) tailing of total RNA prior to reverse transcription ensured that miRNA transcripts would be included in the cDNA product, due to addition of a poly(A) tail to each. Reverse transcription was performed with a modified oligo (dT) primer that included a 5′ universal tag (see reagent and resource table for primers used). This 5′ universal tag enabled mature miRNA-specific primer sets to bind in downstream PCR experiments. Primer sets for different miRNAs were designed in miRprimer software (*Busk, 2014*) with specificity and efficiency was confirmed by dilution series and melt curve analyses, in addition to running the primer sets with cDNA from miRNA mutant samples to confirm a lack of amplification. Both conventional PCR and qPCR were used for miRNA-specific PCR experiments.

## Immunohistochemistry

Dechorionated embryos were fixed in 4% paraformaldehyde for 20 min. Following fixation, embryos were washed 4 X in PBTX (1 X PBS, 0.3% Triton-X) for 15 min at room temperature, prior to the addition of a primary antibody and incubation overnight at 4 °C with gentle rocking. Embryos were washed 4 X in PBTX again prior to incubation with a secondary antibody and DAPI for 2 hr at room temperature (with gentle rocking). Finally, embryos were washed 4 X in PBTX, mounted in 70% glycerol with PBS and stored at 4 °C until imaging.

## Calcium imaging

Calcium imaging was performed using a Leica DM6000 epifluorescence microscope with a 10 X objective. Embryos were aged to 14hAEL, dechorionated and adhered in the ventral-up orientation to a clear glass microscopy slide using Tesa tape glue. Images were captured sequentially using an ET470 40 x ET525 50 m band pass filter for detecting GCaMP6s signal and a ET545 25 x ET605 70 m set for tdTomato, with a single image cycle occurring over 1.5 s. Recording was performed for 6 hr or until

hatching occurred. Fluorescence signals from each channel were measured in FIJI software using ROIs of equal size placed over each embryo. To calculate ΔF/F, the GCaMP6s reading for each frame was divided by the tdTomato reading for the same frame, before subtraction of the baseline calculated as the minimum value of this ratio in a 10-min window centred around each frame.

### Bioinformatic miRNA target prediction

Potential miRNA targets were predicted with the bioinformatic combinatorial target prediction tool ComiR (*Coronnello and Benos, 2013*) that draws upon weighted prediction scores from four major miRNA target prediction algorithms- miRanda, PITA, TargetScan and mirSVR - before integrating them through a machine learning model trained on biochemical data for miRNA-mRNA interactions (*Drosophila* AGO1 IP data – [*Hong et al., 2009*]). From this, a list of predicted targets for a miRNA was generated and ranked by probability score. The following two criteria were applied to filter for targets with a probable role in nervous system functional development: (i) at least one of the following major GO terms: receptor; receptor binding; transporter; small molecule binding; development; nervous system process; behaviour. (ii) Embryonic expression according to modENCODE Development RNA-Seq. A filtered list of the top-10 predicted targets of miR-2b-1, sorted by ComiR score, was subsequently obtained for assessment in biochemical experiments.

### Bioinformatic analysis of CG3638

Evolutionary conservation of *CG3638* protein was determined with PhylomeDB 5 software (*Huerta-Cepas et al., 2014*). For structural analysis, AlphaFold (*Jumper et al., 2021*) software was used to predict protein structure and SACS MEMSAT2 (*Jones et al., 1994*) software was used to visualise transmembrane domains.

### Optogenetic inhibition of neural activity

Embryos were placed on plain 1.5% agar plates and exposed to red light – at 650 nm wavelength and 5000 lux as measured on a EXTECH Instruments 401020 lux meter – for 21 hr. Embryos were subsequently moved to a dark room until hatching, checked regularly under brief weak red light (<1000 lux). Once larvae had hatched, they were moved to a different plan 1.5% agar plate and left for 1 hr under dark conditions. Subsequently, the agar plate was placed on a FIM table and locomotion was tracked for 3 min.

## Acknowledgements

We wish to thank all members of the Alonso Lab for helpful discussions and feedback on this work. We are also grateful to Marta Zlatic for sharing *Drosophila* lines, and to the Bloomington Stock Center and the Vienna *Drosophila* Resource Center for providing fly stocks. This research was funded by a Wellcome Trust Investigator Award (098410/Z/12/Z) made to CRA, a Wellcome Trust Investigator Award (220277/Z20/Z) made to TB and a UK Medical Research Council Project Grant (MR/S011609/1) given to CRA.

## Additional information

### Funding

| Funder | Grant reference number | Author |
| --- | --- | --- |
| Wellcome Trust | 10.35802/098410 | Claudio R Alonso |
| Medical Research Council | MR/S011609/1 | Claudio R Alonso |
| Wellcome Trust | 10.35802/220277 | Tom Baden |

The funders had no role in study design, data collection and interpretation, or the decision to submit the work for publication. For the purpose of Open Access, the authors have applied a CC BY public copyright license to any Author Accepted Manuscript version arising from this submission.

## Author contributions
Jonathan AC Menzies, Formal analysis, Validation, Investigation, Methodology, Writing – original draft, Writing – review and editing; André Maia Chagas, Software, Methodology; Tom Baden, Formal analysis, Methodology; Claudio R Alonso, Conceptualization, Formal analysis, Supervision, Funding acquisition, Investigation, Writing – original draft, Project administration, Writing – review and editing

## Author ORCIDs
Jonathan AC Menzies ⓘ http://orcid.org/0009-0002-2385-8862
André Maia Chagas ⓘ http://orcid.org/0000-0003-2609-3017
Tom Baden ⓘ http://orcid.org/0000-0003-2808-4210
Claudio R Alonso ⓘ http://orcid.org/0000-0001-5761-348X

Reviewer #1 (Public Review) https://doi.org/10.7554/eLife.95209.3.sa1
Reviewer #2 (Public Review) https://doi.org/10.7554/eLife.95209.3.sa2
Author response https://doi.org/10.7554/eLife.95209.3.sa3

---

# Additional files

## Supplementary files
• MDAR checklist

## Data availability
All data generated or analysed during this study are included in the manuscript and supporting files; source data files have been provided for *Figures 1–4* and supplementary figures.

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
