## [Editor Report · eLife assessment]

This **important** study presents a new quantitative imaging pipeline that describes with high temporal precision and throughput the movements of late-stage *Drosophila* embryos, a critical moment when motion first appears. A new approach is used to explore the role of miRNAs in motion onset and presents **solid** evidence that shows a role for miR-2b-1 and its target Motor in embryonic motion. The data are well supported even if the mechanistic insight into the emergence of movement remains to be explored.

---

## [Referee Report · Reviewer #1 (Public Review)]

Summary:

This is an experimentally soundly designed work and a very well-written manuscript. There is a very clear logic that drives the reader from one experiment to the next, the experimental design is clearly explained throughout and the relevance of the acquired data is well analyzed and supports the claims made by the authors. The authors made an evident effort to combine imaging, genetic, and molecular data to describe previously unknown early embryonic movement patterns and to identify regulatory mechanisms that control several aspects of it.

Strengths:

The authors develop a new method to analyze, quantitatively, the onset of movement during the latter embryonic stages of *Drosophila* development. This setup allows for a high throughput analysis of general movement dynamics based on the capture of variations of light intensity reflected by the embryo. This setup is capable of imaging several embryos simultaneously and provides a detailed measure of movement over time, which proves to be very useful for further discoveries in the manuscript. This setup already provides a thorough and quantifiable description of a process that is little known and identifies two different phases during late embryonic movements: a myogenic phase and a neurogenic phase, which they elegantly prove is dependent on neuronal activity by knocking down action potentials across the nervous system.

However, in this system, movement is detected as a whole, and no further description of the type of movement is provided beyond frequency and amplitude; it would be interesting to know from the authors if a more precise description of the movements that take place at this stage can be achieved with this method (e.g. motion patterns across the A-P body axis).

Importantly, this highly quantitative experimental setup is an excellent system for performing screenings of motion regulators during late embryonic development, and its use could be extended to search for different modulators of the process, beyond miRNAs (genetic mutants, drugs, etc.).

Using their newly established motion detection pipeline, the authors identify miR-2b-1 as required for proper larval and embryonic motion, and identify an overall reduction in the quantity of both myogenic and neurogenic movements, as well as an increased frequency in neurogenic movement "pulses".

Focusing on the neurogenic movement phenotype the authors use in situ probes and perform RT-PCR on FACS-sorted CNS cells to unambiguously detect miR-2b-1 expression in the embryonic nervous system. The neurogenic motion defects observed in miR-2b-1 mutant embryos and early larvae can be completely rescued by the expression of ectopic miR-2b-1 specifically in the nervous system, providing solid evidence of the requirement and sufficiency of miR-2b-1 expressed in the nervous system to regulate these phases of movement.

To explore the mechanism through which miR-2b-1 impacts embryonic movement, the authors use a state-of-the-art bioinformatic approach to identify potential targets of miR-2b-1, and find that the expression levels of an uncharacterized gene, CG3638, are indeed regulated by miR-2b-1. Furthermore, they prove that by knocking down the expression of CG3638 in a miR-2b-1 mutant background, the neurogenic embryonic movement defects are rescued, pointing that the repression of CG3638 by miR-2b-1 is necessary for correct motion patterns in wild-type embryos. Therefore, this paper provides the first functional characterization of CG3638, and names this gene Motor.

Finally, the authors aim to discriminate which elements of the embryonic motor system miR-2b-1/Motor are required. Using directed overexpression of miR-2b-1 and Motor knockdown in the motor neurons and the chordotonal (sensory) organs, they prove that the miR-2b-1/Motor regulatory axis is specifically required in the sensory organs to promote normal embryonic and larval movement.

Weaknesses:

The initial screening to identify miRNAs involved in motion behaviors is performed in early larval movement. The logic presented by the authors is clear - it is assumed that early larval movement cannot proceed normally in the absence of previous embryonic motion - and ultimately helped them identify a miRNA required for modulation of embryonic movement. However, it is possible that certain miRNAs play a role in the modulation of embryonic movement while being dispensable for early L1 behaviors. Such regulators might have been missed with the current screening setup.

---

## [Referee Report · Reviewer #2 (Public Review)]

Summary:

The manuscript, ‘A microRNA that controls the emergence of embryonic movement’ by Menzies, Chagas, and Alonso provides evidence that *Drosophila* miR-2b-1 is expressed in neurons and controls the expression of the predicted chloride channel CG3638, here named "Motor". Loss of the miRNA leads to movement phenotypes that can be rescued by downregulation of Motor; using specific drivers, the authors show that a larval movement phenotype (slower movement) can be rescued by knockdown of Motor in the chordotonal organs, suggesting that the increase in Motor found in the chordotonal organs is likely the root of the movement defects. Overall, I found the data presented in the manuscript of reasonable quality and are well enough supported by the presented data.

The genetic and phenotypic analysis seems to be correct. The nicest part of the manuscript is the connection between the loss of a miRNA and finding its likely target in generating a phenotype. The authors also develop some protocols for the analysis of the movement phenotypes which may be useful for others.

---

## [Author Response]

The following is the authors’ response to the original reviews.

**Recommendations for the Authors:**

**Reviewer 1:**
(1) Figure legends are too sparing, and often fail to describe with enough detail and accuracy the experiments presented. Especially in a work like this one, which uses plenty of different approaches and techniques and has a concise main text, description in the figure legends can really help the reader to understand the technical aspects of the experimental design. In my opinion, this will also help highlight the effort the authors put into exploring different and often new technical approaches.

We thank Reviewer 1 for highlighting this point and agree with them that the original figure legends lacked detailed information. In this revised version of our paper we edited all figure legends providing higher detail on experiments and information displayed (see Main text p12-16, Supplementary Information p2-5). We hope this change will improve the clarity and accuracy of the description of our experiments.

**Reviewer 2:**
(1) Is there evidence that the early movement phenotype is actually linked to the larval movement phenotype? I noticed that the chordotonal driver experiment was only examined for larval movement. Is this driver not expressed earlier? Could the authors check the early phenotype using this driver? Are there early drivers that are expressed in chordotonal organ precursors (not panneuronal) and does the knockdown of CG3638 in these specific cells suppress the early phenotype?(2) More broadly, I would like to understand the function of the early embryonic movements. My concern is that they may only be a sign that the nervous system is firing up. If the rescue of the late miRNA mutant phenotype with chordotonal organ expression is only through a late change in the expression of CG3638, then the larval phenotype is probably not due to a developmental change, but a change in the immediate functioning of the neurons. Would this suggest that the early pulsing is not required for anything, at least at our level of understanding? If the driver is actually expressed early and late, then perhaps the authors could test later drivers to delimit the early and late functions of the miRNA?

The comments by Reviewer 2 in the points above are important and enquire about the biological role of early embryonic movements and whether these movements are linked to later larval activity or are somewhat irrelevant to the behaviour of the animal at later stages.

To address this important question, we conducted a new experiment in which we reduced neural activity specifically in the embryo (i.e. from 10hs AEL until the end of embryogenesis) and tested whether this treatment had any impact on larval movement. If – as put by Rev2 – the ‘early pulsing is not required for anything’ and the larval phenotype emerges from an acute change in neuronal physiology, then our experiment should show no effects at the larval stage. The results shown in Figure S4 (see Supplementary Information, p5) show that this is not the case: artificial reduction of neural activity during embryogenesis leads to a statistically significant reduction in larval speed, similar to that caused by the loss of miR-2b-1. This shows that modifications of embryonic activity impact larval movement.

Furthermore, earlier work on the biological role of embryonic activity identified an activity-dependent ‘critical period’ during late embryogenesis (Giachello and Baines, 2015; Ackerman et al., 2021): manipulations at or around this critical period result in both locomotor and seizure phenotypes in larvae. We cite these papers in the main text (p7).

In addition, two recent papers (Zeng et al., 2021; Carreira-Rosario et al., 2021) – which we cite in the main text (p5) – show that inhibition of muscle activity specifically during the embryonic period prevents the generation of normal neural activity patterns in both, embryo and larva. Similar results are observed when proprioceptive sensory inputs to the central nervous system are blocked, with larval locomotion also disrupted.

Altogether, the data already in the literature plus our new addition to the paper, show that early embryonic movements play a key role in the development of the nervous system and larval locomotion.

(3) Given the role in the larval chordotonal organs, have the authors also checked the adult movements?

The question of whether miR-2b-1 action in chordotonal organs affects behaviour at later stages of the *Drosophila* life cycle is interesting and was the reason why we assessed different genetic manipulations at the larval stage. However, we believe that assessing adult locomotor phenotypes is beyond the scope of this paper.

(4) The authors state that mir-2b-1 is a mirtron. I do not believe this is correct. It is not present in an intron in Btk from what I can see. Also, in the reference that the authors use when stating that mir-2b is a mirtron, I believe mir-2b-1 is actually used as a non-mirtron control miRNA. As mirtrons are processed slightly differently from regular hairpins and often use only the 3' end of the hairpin for miRNA creation, this may not be a trivial distinction.

We are grateful to Rev2 for highlighting this point: indeed, as they say, miR-2b-1 is located in the 3’UTR of host gene Btk, rather than in an intron. Accordingly, in this revision we remove the comment on miR-2b-1 being a mirtron (p6) and deleted the citation.

(5) For miRNA detection, the authors use in situ hybridization and QPCR. Both methods show that the gene is expressed but not that the mature miRNA is made. If the authors wanted a truly independent test for the presence of the miRNA, a miRNA sensor might be a better choice and it would hint at which part of the hairpin makes the functional miRNA. This is probably not necessary but could be a nice addition.

We thank Rev2 for drawing attention to this point and allowing this clarification. The qPCR protocol we used is based on the method developed by Balcells et al., 2011 (w/303 citations) (see Materials and Methods section in Supplementary Information, p14) which allows the specific amplification of mature miRNA transcripts, and not their precursors. This method for mature miRNA PCR is so robust that it has even been patented (WO2010085966A2). To ensure that the reader is clear about our methods, we state in the main text (p6) that we perform "RT-PCR for the mature miRNA transcript". [NB: miRNA sensors provide a useful method to assess miRNA expression but can also act as competitive inhibitors of physiological miRNA functions, titrating away miRNA molecules from their real targets in tissue; therefore, results using this method are often difficult to interpret.]

(6) Curious about mir-2b-1 and any overlap with the related mir2b-2 and the mir2a genes. I am just wondering about the similarity in their sequences/targets and if they might have similar phenotypes or enhance the phenotypes being scored by the authors.

This is an interesting point raised by REV2 and indeed miR-2b-1 does belong to the largest family of microRNAs in *Drosophila*, the miR-2 family, discussed in detail by Marco et al., 2012. However, we consider that performing tests of additional miRNA mutations, both individually and in combination with miR-2b-1, is beyond the scope of this paper.

(7) Related to this, the authors show that the reduction of a single miRNA target suppresses the miRNA loss of function phenotype. This indicates that this target is quite important for this miRNA. I wonder if the target site is conserved in the human gene that the authors highlight.

This is another interesting comment by Rev2. To pursue their idea, we have performed a blast for the miR-2b-1 target site in the human orthologs of CG3638 and did not find a match suggesting that the relationship between miR-2b-1 and CG3638 is not evolutionarily preserved between insects and mammals.

**Public Reviews:**

**Reviewer #1:**
Weaknesses:The authors do not describe properly how the miRNA screening was performed and just claim that only miR-2b-1 mutants presented a defective motion phenotype in early L1. How many miRNAs were tested, and how candidates were selected is never explicitly mentioned in the text or the Methods section.

We identified miR-2b-1 as part of a genetic screen aimed at detecting miRNAs with impact on embryonic movement, but this full screen is not yet complete. Seeing the clear phenotype of miR2b-1 in the embryo prompted us to study this miRNA in detail, which is what we report in this paper.

The initial screening to identify miRNAs involved in motion behaviors is performed in early larval movement. The logic presented by the authors is clear - it is assumed that early larval movement cannot proceed normally in the absence of previous embryonic motion - and ultimately helped them identify a miRNA required for modulation of embryonic movement. However, it is possible that certain miRNAs play a role in the modulation of embryonic movement while being dispensable for early L1 behaviors. Such regulators might have been missed with the current screening setup. Although similar changes to those described for the neurogenic phase of embryonic movement are described for the myogenic phase in miR-2b-1 mutants (reduction in motion amplitude), this phenotype goes unexplored. This is not a big issue, as the authors convincingly demonstrate later that miR-2b-1 is specifically required in the nervous system for proper embryonic and larval movement, and the effects of miR-2b-1 on myogenic movement might as well be the focus of future work. However, it will be interesting to discuss here the implications of a reduced myogenic movement phase, especially as miR-2b-1 is specifically involved in regulating the activity of the chordotonal system - which precisely detects early myogenic movements.

We thank Rev1 for their interest in that loss of miR-2b-1 results in a decrease in movement during the myogenic phase, in addition to the neurogenic phase. Indeed, two recent papers (Zeng et al., 2021; Carreira-Rosario et al., 2021) – which we cite in the main text (p5) – show that inhibition of muscle activity during a period that overlaps with the myogenic phase prevents the formation of normal neural activity patterns and larval locomotion. They also observe the same when inhibiting proprioceptive sensory inputs to the central nervous system. This could suggest that the effects of miR-2b-1 on the myogenic phase might have ‘knock-on’ effects upon the later neurogenic phase and larval movement. However, we note that genetic restoration of miR-2b-1 expression specifically to neurons completely rescues the larval speed phenotype (Fig. 3G), suggesting that the dominant effect of miR-2b-1 upon movements is through its action within neurons. To recognise Rev1’s comment we have added a short sentence to the text (p7) suggesting that ‘the effects of miR-2b-1 observed at earlier stages (myogenic phase) are possibly offset by normal neural expression of miR-2b-1’.

FACS-sorting of neuronal cells followed by RT-PCR convincingly detects the presence of miR-2b-1 in the embryonic CNS. However, control of non-neuronal cells would be required to explore whether miR-2b-1 is not only present but enriched in the nervous system compared to other tissues. This is also the case in the miR-2b-1 and Janus expression analysis in the chordotonal organs: a control sample from the motor neurons would help discriminate whether miR-2b-1/Janus regulatory axis is specifically enriched in chordotonal organs or whether both genes are expressed throughout the CNS but operate under a different regulation or requirements for the movement phenotypes.

The RNA in situ hybridisation data included in the paper (Fig. 3B) show that RNA probes for miR2b-1 precursors reveal very strong signal in neural tissue – with very low signal detected in other tissues – strongly indicating that expression of miR-2b-1 is highly enriched in the nervous system.

**Reviewer #2:**
Weaknesses:As I mentioned above, I felt the presentation was a bit overstated. The authors present their data in a way that focuses on movement, the emergence of movement, and how their miRNA of interest is at the center of this topic. I only point to the title and name that they wish to give the target of their miRNA to emphasize this point. "Janus" the GOD of movement and change. The results and discussion section starts with a paragraph saying, "Movement is the main output of the nervous system... how developing embryos manage to organise the necessary molecular, cellular, and physiological processes to initiate patterned movement is still unknown. Although it is clear that the genetic system plays a role, how genes control the formation, maturation and function of the cellular networks underlying the emergence of motor control remains poorly understood." While there is nothing inherently untrue about these statements, it is a question of levels of understanding. One can always argue that something in biology is still unknown at a certain level. However, one could also argue that much is known about the molecular nature of movement. Next, I am not sure how much this work impacts the area of study regarding the emergence of movement. The authors show that a reduction of a miRNA can affect something about certain neurons, that affects movement. The early movements, although slightly diminished, still emerge. Thus, their work only suggests that the function of some neurons, or perhaps the development of these neurons may impact the early movements. This is not new as it was known already from early work from the Bate lab. Later larval movements were also shown to be modified in the miRNA mutants and were traced to "janus" overexpression in the chordotonal organs. As neurons are quite sensitive to the levels of Cl- and Janus is thought to be a Cl- channel, this could lead to a slight dysfunction of the chordotonal neurons. So, based on this, the work suggests that dysfunction of the chordotonal organs could impact larval movement. This was, of course, already known. The novelty of this work is in the genes being studied (important or not). We now know that miR 2b-1 and Janus are expressed in the early neurons and larval chordotonal neurons and their removal is consistent with a role for these genes in the functioning of these neurons. This is not to trivialize these findings, simply to state that these results are not significantly changing our overall understanding of movement and the emergence of movement. I would call it a stretch to say that this miRNA CONTROLS the emergence of movement, as in the title.

As already mentioned in our provisional response, on this point we politely – but strongly – disagree with Rev2’s suggestion that the findings are inflated by our language. We also note that they criticise our use of the verb ‘control’, yet this is a standard textbook term in molecular biology to describe biological processes regulated by genetic factors: given that miR-2b-1 regulates movement patterns during embryogenesis, to say that miR-2b-1 ‘controls’ embryonic movement in the *Drosophila* embryo is reasonable and in line with the language used in the field.

Finally, the name Janus should be changed as it is already being used. A quick scan of flybase shows that there is a Janus A and B in flies (phosphatases) and I am surprised the authors did not check this. I was initially worried about the Janus kinase (JAK) when I performed the search. While I understand that none are only called Janus, studies of the jan A and B genes refer to the locus as the janus region, which could lead to confusion. The completely different molecular functions of the genes relative to CG3638 add to the confusion. Thus, I ask that the authors change the name of CG3638 to something else.

Thank you for spotting this omission. In the revised MS we propose a new name – Movement Modulator (Motor) – for the gene previously described as Janus (CG3638) to avoid annotation issues at FlyBase due to other, unrelated genes that include this word as part of their names. All instances where Janus was used are now replaced by Motor (abstract; main text pages 9-10; Figure 4).